# ENHANCING LLMS FOR KNOWLEDGE BASE QUESTION ANSWERING BY CHAIN-OF-DECOMPOSITION

**Yonggang Zhang**
Hong Kong University of Science and Technology
zhangyg@ust.hk

**Jianqi Gao**
Shanghai University
jianqi_gao@shu.edu.cn

**Jie Lu**
University of Technology Sydney
jie.lu@uts.edu.au

## ABSTRACT

Large language models (LLMs) have demonstrated remarkable success across diverse domains through in-context learning or fine-tuning. However, adapting LLMs to Knowledge Base Question Answering (KBQA) remains challenging, as KBQA necessitates multi-step reasoning over large-scale structured knowledge bases. Directly prompting LLMs with entire knowledge bases incurs prohibitive computational costs, while existing methods provide limited guidance on effectively fine-tuning LLMs for such complex reasoning tasks. In this work, we propose Chain-of-Decomposition (`CoD`), a novel framework that decomposes KBQA into three modular steps: (1) an LLM-free retrieval module to extract query-relevant subgraphs from the knowledge base, (2) a parameter-free reformulation step that transforms retrieved contexts into structured reasoning paths, and (3) a lightweight LLM-based reasoning module trained to evaluate the logical validity of each path. By isolating computation-heavy retrieval and rule-based reformulation from LLM reasoning, `CoD` reduces task complexity and enables efficient fine-tuning focused solely on the final verification step. Comprehensive experiments demonstrate that Llama-2 7B, fine-tuned with the proposed `CoD` surpasses strong baselines, including GPT-4 augmented with retrieved knowledge, achieving state-of-the-art performance on WebQSP and CWQ benchmarks. Our code is publicly available at https://github.com/YonggangZhang9412/KBQA-CoD.

## 1 INTRODUCTION

The great achievement of large language models (LLMs) have attracted widespread attention (Brown et al., 2020; Ouyang et al., 2022b; OpenAI, 2023). The striking feature of LLMs is their ability to handle complex tasks through reasoning (Wei et al., 2022; Wang et al., 2023b). Despite their impressive performance, LLMs have substantial limitations when facing complex knowledge reasoning tasks (Pan et al., 2024) that require multi-hop reasoning. In this regard, advanced works propose to adapt LLMs to these complex tasks by either prompting LLMs with knowledge (Sun et al., 2023; Sui et al., 2024) or fine-tuning LLMs with carefully designed data and objectives (Luo et al., 2023b).

However, it is challenging to adapt LLMs to the tasks of knowledge base question answering (KBQA). This results from the fact that KBQA aims to reason about answers for an input query, which requires a large-scale knowledge base and deep reasoning. Specifically, prompting LLMs with such a large-scale knowledge base in an in-context learning manner leads to high computation costs. Meanwhile, fine-tuning LLMs to promote the reasoning capability relies heavily on the utilized training data and objective functions (Ouyang et al., 2022a). In this regard, some works propose to scale down the size of the knowledge base by retrieving query-relevant context (Sun et al., 2023; Sui et al., 2024). Advanced works propose to fine-tune LLMs with appropriate objective functions (Luo et al., 2023a;b), aiming to encourage LLMs to predict the answer directly or indirectly. However, the existing works mainly focus on improving the capability of LLMs for complex reasoning, while overlooking the strategy to reduce the task complexity.

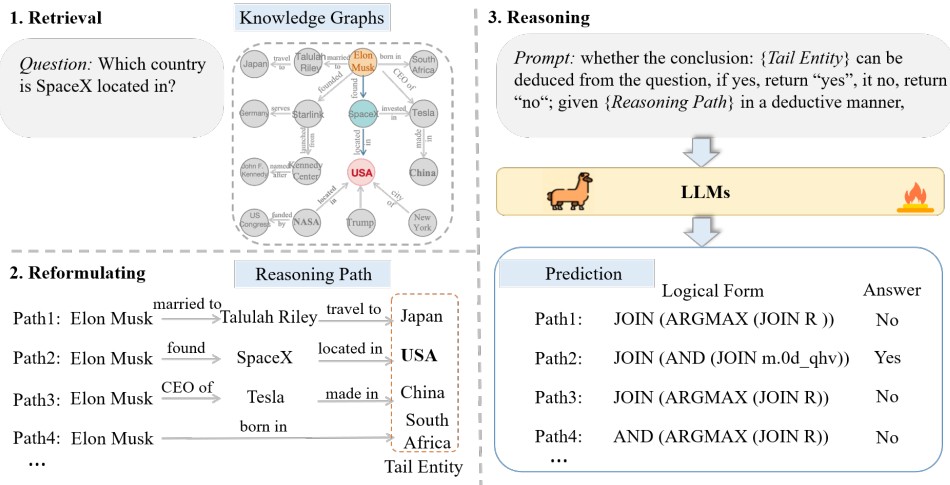

Figure 1: CoD overview: Query-relevant information is first retrieved. Reasoning paths are then constructed. Finally, LLMs are fine-tuned to determine whether reasoning paths are logically valid.

In this work, we propose Chain-of-Decomposition (CoD), a novel framework that simplifies the task by decomposing KBQA into three modular steps. As shown in Fig. 1 and Eq.(5), we decompose the KBQA tasks into a chain with three subtasks: retrieval, reformulating, and reasoning by factoring the distribution of the answer generation. The factorization is based on our constructed causal graph (Fig. 2), which formulates the answer-generation process. First, the objective of the retrieval task is to scale down the knowledge base by returning query-relevant information, which is an LLM-free retrieval module to extract query-relevant subgraphs from the knowledge base. This leads to a simple binary task problem[1]. Second, the objective of the reformulating task is to transform the retrieved context into reasoning paths, which is a parameter-free reformulation step. Namely, we can perform the reasoning path construction by designing some rules, leading to no learnable parameters for the task. Ultimately, the objective of the reasoning task is to determine whether a reasoning path is logically valid, which is a lightweight LLM-based reasoning module. Through the decomposition, we surprisingly find that two of these tasks are LLM-free, which isolates computation-heavy retrieval and rule-based reformulation from LLM reasoning. Consequently, our decomposition provides detailed instructions on how to simplify KBQA tasks and fine-tune LLMs for only one task, which drastically reduces task complexity and enables efficient fine-tuning of LLMs.

To verify the efficacy of the proposed CoD, we conduct experiments on two benchmark KBQA datasets: WebQuestionSP (WebQSP) (Yih et al., 2016) and ComplexWebQuestions (CWQ) (Talmor & Berant, 2018). Comprehensive results demonstrate that fine-tuning Llama-2 7B with CoD can outperform all baseline methods by a considerable margin, achieving state-of-the-art (SOTA) performance. Moreover, the fine-tuned Llama-2 7B outperforms the GPT4 model with retrieved knowledge.

Our main contributions can be summarized as follows.

- We construct a causal graph to formulate the answer-generation process in KBQA. Based on the graph, we factorize the distribution of answer prediction, which motivates a decomposition strategy to reduce the complexity of the KBQA task.

- We propose chain-of-decomposition (CoD), a novel framework to adapt LLMs to KBQA by decomposing the task into three modular steps: retrieval, reformulating, and reasoning. CoD promotes LLM-based KBQA by isolating retrieval and reformulation from reasoning, resulting in a simple reasoning task for LLMs.

- The proposed method CoD dramatically outperform the existing methods, achieving state-of-the-art (SOTA) performance. Moreover, fine-tuning Llama-2 7B with CoD outperforms

---

[1]Our experimental results (Table 2) demonstrate that employing a relatively small generative model, i.e., T5, can achieve exciting performance.

the GPT4 model with retrieved knowledge, highlighting its ability to empower lightweight LLMs for complex reasoning.

## 2 PRELIMINARY

### 2.1 KNOWLEDGE BASE QUESTION ANSWERING

Knowledge base question answering (KBQA) is a reasoning task based on a large-scale structured knowledge base, i.e., knowledge graph (KG). The KG $\mathcal{G}$ is composed of factual knowledge in the form of a set of triples: $\mathcal{G} = \{(e_s, r, e_o) | e_s, e_o \in \mathcal{E}, r \in \mathcal{R}\}$, where $e_s$ and $e_o$ are the *s*ubject entity and *o*bject entity connected by the relation $r$.

The goal of the KBQA task is to construct a function $f$ to predict answers $A \in \mathcal{A}(Q)$ for an input query $Q$ in the form of natural language question based on the knowledge from $\mathcal{G}$, i.e., $\hat{A} = f(Q, \mathcal{G})$. Here, we leverage $\mathcal{A}(Q)$ to represent the fact that the set of answers varies with the input query $Q$. In this work, we follow previous work (Jiang et al., 2023) and merely consider the close-set scenarios where the topic entities $\mathcal{T}(Q)$ used in $Q$ and the answer entities $\mathcal{A}(Q)$ used in $A$ are involved in $\mathcal{G}$, e.g., $\mathcal{T}(Q), \mathcal{A}(Q) \subseteq \mathcal{E}$. More details can be found in Appendix A.

### 2.2 TWO STRATEGIES FOR KBQA

There are mainly two types of approaches to realize the predictor $f(Q, \mathcal{G})$. The first approach is straightforward, realizing $f$ by directly generating answers,

$$\min_{\theta} KL(P(A|Q, \mathcal{G}), f_{\theta}(A|Q, \mathcal{G})), \tag{1}$$

where $A$ and $Q$ denote the answer and query random variables, $P(A|Q, \mathcal{G})$ represents the target distribution of answer generation, $\theta$ denotes the parameter of the predictor $f$. Given the objective function in Eq. (1), we can complete the KBQA task by converting it to either a classification problem (Sun et al., 2019) or a generation problem (Saxena et al., 2022). Note that the used KG is typically large. Thus, the prediction is usually based on a retrieved subgraph relevant to the query.

Different from the approach of directly generating answers, semantic parsing-based methods parse the input query into a logical form (LF), which is then executed over $\mathcal{G}$ to obtain the answers. This can be formalized as follows,

$$\min_{\theta} KL(P(M|Q, \mathcal{G}), g_{\theta}(M|Q, \mathcal{G})), \tag{2}$$

where $M$ indicates the LF, $P(M|Q, \mathcal{G})$ is the distribution of LF generation, $g$ is the function designed to generate LF parameterized by $\theta$. In this regard, the answer is generated by executing the SPARQL query converted from the predicted LF $\hat{M} = g_{\hat{\theta}}(M|Q, \mathcal{G})$ against the KG,

$$\hat{a} = \texttt{Execute}(\texttt{Convert}(\hat{M})), \tag{3}$$

where $\hat{\theta}$ denotes the optimized parameter of $g$, $\texttt{Execute}(\cdot)$ is the query execution function, and $\texttt{Convert}(\cdot)$ represents the conversion function transforming LF $\hat{M}$ to SPARQL query. Similar to the direct generation approach, the semantic parsing approach usually predicts with a retrieved subgraph relevant to the query as the used KG is large (Yu et al., 2022; Luo et al., 2023b).

Given a query $Q$, these two strategies retrieve question-relevant context $C$, leading to a set of reasoning paths $\mathcal{Z}$. Subsequently, the answer $A$ can be directly ($\mathcal{Z}/C \longrightarrow a$) or indirectly ($\mathcal{Z}/C \longrightarrow m \longrightarrow a$) produced with these reasoning paths or context. Here, the reasoning path $z \in \mathcal{Z}$ is formatted as a series of directed relations and the corresponding entities: $z = e_0 \xrightarrow{r_1} e_1 \xrightarrow{r_2}, \ldots, \xrightarrow{r_k} e_k$ with the $i$-th entity $e_i \in \mathcal{E}$ and the $i$-th relation $r_i \in \mathcal{R}$. For instance, $z = $ The Baltimore Fight Song $\xrightarrow{\texttt{fight\_song}}$ Baltimore Ravens $\xrightarrow{\texttt{championships}}$ Super Bowl XXXV, Super Bowl XLVII means that "Baltimore Fight Song" is the fight song of the team "Baltimore Ravens", and "Baltimore Ravens" is the champion of "Super Bowl XXXV" and "Super Bowl XLVII".

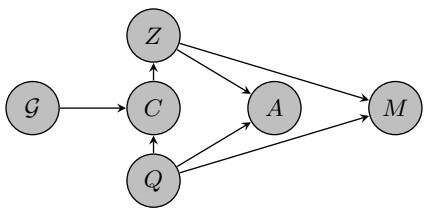

Figure 2: Causal graph of the answer generation process in KBQA. Each node represents a random observable, where $\mathcal{G}, Q, C, Z, A, M$ are knowledge graph, question, context, reasoning path, answer, and logical form, respectively.

## 3 METHOD

### 3.1 MOTIVATION

The KBQA task is challenging for LLMs (Luo et al., 2023a;b) due to the fact that completing KBQA requires complex reasoning. In this regard, one promising approach is to fine-tune LLMs to promote their performance. However, fine-tuning LLMs with limited data to accomplish a complex task would be challenging. To this end, we revisit the processes widely adopted to accomplish KBQA tasks, aiming to simplify the KBQA task for LLM fine-tuning.

**For direct answer-generation approach**, we can directly generate the answer by training a predictor $f_\theta(A|Q, \mathcal{G})$ to approach the target distribution $P(A|Q, \mathcal{G})$ using the objective function in Eq. (1). As it is challenging to fine-tune LLMs with limited data for complex tasks, we aim to simplify the task. In this regard, the factorization approach used in causal inference provides a straightforward approach (Peters et al., 2017). The answer-generation process can be formalized by a causal graph (Peters et al., 2017), as depicted in Figure 2. Based on the graph, we can factorize the probability of generating answers $P(A|Q, \mathcal{G})$ using the independence between variables. Given the causal graph formalizing the answer-generation process, we factorize the joint probability of all variables as follows,

$$P(Q, \mathcal{G}, C, Z, A) = P(Q)P(\mathcal{G})P(C|Q, \mathcal{G})P(Z|C)P(A|Q, Z), \tag{4}$$

where the input query $Q$ and the knowledge base $\mathcal{G}$ are the causes of the context $C \subset \mathcal{G}$, $P(C|Q, \mathcal{G})$ denotes the mechanism of retrieving question-relevant context over the knowledge base, $P(Z|C)$ is the mechanism to generate reasoning paths $Z$, and $P(A|Q, Z)$ stands for the mechanism of reasoning the answer $A$ based on the question and reasoning paths. Note that we omit the logical form random variable $M$ when considering the direct answer generation approach.

According to the factorization above and using the chain rule of probabilities, we can rewrite the probability of generating answer $P(A|Q, \mathcal{G})$ as follows[2],

$$P(A|Q, \mathcal{G}) = \sum_Z \underbrace{P(A|Q, Z)}_{\text{reason}} \sum_C \underbrace{P(Z|C)}_{\text{reformulate}} \underbrace{P(C|Q, \mathcal{G})}_{\text{retrieve}}, \tag{5}$$

where we marginalize over the context $C$ and the reasoning path $Z$. The factorization is intuitive. Specifically, we can ***reason*** about the answer using the query and reasoning path, i.e., $P(A|Q, Z)$. This is based on the fact that we can ***reformulate*** the retrieved context into a specific type of reasoning path using $P(Z|C)$. Here, the premise is that we can ***retrieve*** query-relevant context from the knowledge base using the input query, i.e., $P(C|Q, \mathcal{G})$.

**For logical form-generation approach**, we can decompose the logical form (LF)-generation process using the causal graph in Figure 2 by omitting the answer $A$. Based on the graph, we can factorize $P(M|Q, \mathcal{G})$, i.e., the probability of generating LFs. To this end, we first factorize the joint probability of all variables as follows,

$$P(Q, \mathcal{G}, C, Z, M) = P(Q)P(\mathcal{G})P(C|Q, \mathcal{G})P(Z|C)P(M|Q, Z), \tag{6}$$

where the mechanism of retrieving question-relevant context over the knowledge base $P(C|Q, \mathcal{G})$ and the mechanism to generate reasoning paths $Z$ using the context $P(Z|C)$ are the same as those

---

[2]Here, we consider the context and question as discrete variables.

under the direct answer-generation approach, and $P(M|Q, Z)$ stands for the mechanism of reasoning the LF $M$ based on the question and reasoning paths. Then, we have an intuitive decomposition

$$P(M|Q, \mathcal{G}) = \sum_R \underbrace{P(M|Q, Z)}_{\text{reason}} \sum_C \underbrace{P(Z|C)}_{\text{reformulate}} \underbrace{P(C|Q, \mathcal{G})}_{\text{retrieve}}. \tag{7}$$

According to Eq. (7), we can **reason** about LFs using $P(M|Q, Z)$, where we **reformulate** the retrieved context into reasoning paths using $P(Z|C)$. Here, the premise is that we can **retrieve** query-relevant context using $P(C|Q, \mathcal{G})$.

Factorization provides an alternative approach to decomposing complex KBQA tasks into simple subtasks. Namely, we can decompose the objectives of these two strategies into three subtasks: reasoning, reformulating, and retrieving. Moreover, the objectives of reformulating and retrieving are the same for the direct answer-generation approach and the logical-form generation approach.

## 3.2 Objective of Reasoning Model

LLMs have achieved exciting success in various reasoning tasks. Thus, we employ an open-source LLM to realize the reasoning model following previous work (Luo et al., 2023a;b). Inspired by previous work (Yu et al., 2022), we encourage LLMs to generate answers and logical forms simultaneously. This also aligns well with our factorization perspective, as these two approaches share the same two sub-objectives, i.e., retrieving and reformulating.

According to Eq. (5), we aim to construct a reasoning model to approach $P(A|Q, Z)$ by

$$\min_{\theta_r} KL(P(A|Q, Z), f_{\theta_r}(A|Q, Z)), \tag{8}$$

where $\theta_r$ represents the learnable parameter of the reasoning model $f_{\theta_r}$ to generate answers. Note that the number of the candidate values of $A$ is large, making the task challenging. In this regard, we propose to transform the multiple-class classification task into a binary classification task by grouping answer candidates/entities into one class with the rest of the candidates/entities into the other class. This transforms the original answer $A \in \mathcal{A}$ to the label $Y_a(Q, Z) \in \{0, 1\}$, where $Y_a(Q, Z) = 1$ represents that the input pair $(Q, Z)$ can derive the correct answer $A$ and $Y_a(Q, Z) = 0$ means the failure of deriving the correct answer. We can rewrite the objective function in Eq. (8) as follows,

$$\min_{\theta_r} L_a(\theta_r) \triangleq CE(f_{\theta_r}(A|Q, Z), Y_a(Q, Z)), \tag{9}$$

where $CE(\cdot, \cdot)$ denotes the cross-entropy loss. As LLMs are trained for generation tasks rather than classification tasks, we map the label to the token space, i.e., $Y_a(Q, Z) = 1$ is labeled as "yes" and "no" for $Y_a(Q, Z) = 0$.

For LF-generation approach, according to Eq. (7), we aim to construct a reasoning model to estimate $P(M|Q, Z)$ by

$$\min_{\theta_r} KL(P(M|Q, Z), f_{\theta_r}(M|Q, Z)), \tag{10}$$

where $\theta_r$ represents the learnable parameter of the reasoning model $f_{\theta_r}$ to generate LFs. To fine-tune LLMs, we realize the objective in Eq. (10) using the widely adopted objective of next-token prediction with LF as the ground-truth label,

$$\min_{\theta_r} L_m(\theta_r) \triangleq -\log \prod_{i=1}^{|M|} f_{\theta_r}(M_i|Q, Z, M_{0:i-1}), \tag{11}$$

where we use $M$ to represent $M(Q, Z)$ for brevity and $M_i$ denotes the $i$-th token in the LF $M$. Thus, we can fine-tune LLMs to construct a reasoning model using the following objective,

$$\min_{\theta_r} L_r(\theta_r) \triangleq L_a(\theta_r) + L_m(\theta_r). \tag{12}$$

## 3.3 Objective of the Reformulating Model

According to Eq. (5), we aim to reformulate the context into reasoning paths to approach $P(Z|C)$ by

$$\min_w KL(P(Z|C), h_w(Z|C)), \tag{13}$$

where $C$ denotes the retrieved context or subgraph and $h_w$ is the reformulating model parameterized with $w$. In KBQA, reformulating context into a reasoning path can be realized in a handcrafted manner. Namely, we can realize $P(Z|C)$ in a parameter-free scheme by designing a rule of transformation. For instance, we can design a template to reformulate the given context into a reasoning path. Consequently, we have $KL(P(Z|C), h_w(Z|C)) = 0$, as $w = \emptyset$ is an optimal solution. Thus, to enable the translation of context into reasoning paths, we propose a deterministic algorithm that operates without necessitating model training.

For instance, we can reformulate the context as follows (more details about the reasoning path template can be found in Appendix B.). Note that not all reasoning paths can lead to the answer. In the case of dense subgraphs, the reformulation step filters structurally invalid path combinations, reducing the exponential search space to linearly structured chains. For sparse subgraphs, imposing connectivity constraints through path construction still yields significant gains over directly feeding raw triples to the LLM, as confirmed by the ablation in Table 3 (+14.33% improvement from the 'Only RP' variant).

---

**Retrieved context:**
(*Penn State, contained_by, State College*),  (*Penn State, contains, Beaver Stadium*),
(*Penn State, colors, Royal blue and white*),  (*State College, contained_by, Pennsylvania*),
(*Beaver Stadium, contained_by, University Park*).

**Reasoning paths:**
$z_1 = $ Penn State $\xrightarrow{\texttt{contained\_by}}$ State College $\xrightarrow{\texttt{contained\_by}}$ Pennsylvania;
$z_2 = $ Penn State $\xrightarrow{\texttt{contains}}$ Beaver Stadium $\xrightarrow{\texttt{contained\_by}}$ University Park.

---

### 3.4 OBJECTIVE OF THE RETRIEVING MODEL

According to Eq. (5), we aim construct a retriever to approach $P(C|Q, \mathcal{G})$ by

$$\min_{\theta_e} KL(P(C|Q, \mathcal{G}), \phi_{\theta_e}(C|Q, \mathcal{G})), \tag{14}$$

where $C$ denotes a subgraph of the structured knowledge base and $\phi_{\theta_e}$ is the retrieving model parameterized with $\theta_e$. The computational cost of traversing all subgraphs of a large-scale knowledge base is prohibitive. Thus, we follow previous works to shrink the number of candidate contexts by filtering out entities and their corresponding relations that lead to a large hop count (or path length). This results in a small-scale graph $\mathcal{G}_s(Q)$, where $Q$ represents the set constructed for the question $Q$.

Although filtering out a large number of entities and tuples leads to a small-scale graph $\mathcal{G}(Q)$, the number of the rest of the entities could be large. Moreover, the small-scale graph $\mathcal{G}(Q)$ varies with the query $Q$. Thus, it is challenging to train a classifier. In this regard, we propose to divide relations $O$ in $\mathcal{G}(Q)$ into two classes. Namely, we merge all answer-relevant relations into a single set[3] and merge the remaining relations into another class. Consequently, we can retrieve answer-relevant context by determining whether a relation is answer-relevant. Note that the retrieved relation and its corresponding subject and object entities are used to construct the set of answer-relevant contexts. Thus, we can rewrite the objective of the retrieving model

$$\min_{\theta_e} CE(\phi_{\theta_e}(Q, \mathcal{G}_s(q), O), Y_e(Q, O)), \tag{15}$$

where $Y_e(Q, O) \in \{0, 1\}$ stands for the label of the relation $O$. Namely, $Y_e(Q, O) = 1$ represents that the relation $O$ is answer-relevant, while $Y_e(Q, O) = 0$ means that the relation will not be included in the context. The retrieving task is a traditional binary classification task. Thus, we propose to train a relatively small model, e.g., T5, rather than fine-tuning LLMs to construct the retrieving model.

### 3.5 DISCUSSIONS

We find that RoG (Luo et al., 2023b) also formally employs the concept of reasoning path, which is regarded as a natural baseline of our method. Mathematically, by assuming that the generation of the answer $A$ to the question $Q$ is independent of the question-relevant context $C$ (**Assumption A1**)

---

[3]For entity in the query, retrieval is completed by determining whether a relation should be involved.

Table 1: Performance comparison of `CoD` with different baselines on the two KBQA datasets.

| Backend Models | Model | WQSP | | CWQ | |
|---|---|---|---|---|---|
| | | Hits@1(%) | F1(%) | Hits@1(%) | F1(%) |
| Non-LLM | TIARA (Shu et al., 2022) | 75.20 | 78.90 | - | - |
| | UniKGQA (Jiang et al., 2023) | 77.20 | 72.20 | 51.20 | 49.40 |
| | HGNet (Chen et al., 2022) | 76.90 | 76.60 | 68.90 | 68.50 |
| Prompting - LLM Only (GPT-3.5) | Zero-shot | 54.37 | 52.31 | 34.87 | 28.32 |
| | Few-shot | 56.33 | 53.12 | 43.21 | 35.85 |
| | CoT (Wei et al., 2022) | 57.42 | 54.72 | 43.21 | 35.85 |
| Prompting - LLM Only (GPT-4) | Zero-shot | 62.32 | 59.71 | 42.71 | 37.93 |
| | Few-shot | 68.85 | 62.71 | 51.52 | 43.70 |
| | CoT (Wei et al., 2022) | 72.11 | 65.37 | 53.51 | 44.76 |
| Prompting - LLM + KG (GPT-3.5) | ToG (Sun et al., 2023) | 75.13 | 72.32 | 57.59 | 56.96 |
| | FiDeLis(Sui et al., 2024) | 79.32 | 76.78 | 63.12 | 61.78 |
| Prompting - LLM + KG (GPT-4) | ToG (Sun et al., 2023) | 81.84 | 75.97 | 68.51 | 60.20 |
| | FiDeLis(Sui et al., 2024) | 84.39 | 78.32 | 71.47 | 64.32 |
| Finetuning - LLM + KG | DeCAF (Yu et al., 2022) | 82.10 | - | 70.42 | - |
| | KD-CoT (Wang et al., 2023a) | 73.70 | 50.20 | 50.50 | - |
| | RoG (Luo et al., 2023b) | 83.15 | 69.81 | 61.39 | 56.17 |
| | CoD (ours) | **86.54** | **81.24** | **77.42** | **65.70** |

and the generation of reasoning paths $Z$ is independent of the knowledge base $\mathcal{G}$ (**Assumption A2**), we show in Appendix C that our method intrinsically subsumes RoG as a special exemplar of itself[4]. Methodologically, in contrast to RoG which solves the optimization problem of Eq. (1) under the EM framework, where a rigorous uniform assumption is explicitly imposed, our method, thanks to the reformulation of $P(A|Q, \mathcal{G})$ in Eqs. (5,7), can address the optimization problem of Eq. (1) by aligning each subterm in $P(A|Q, \mathcal{G})$ and $f_\theta(A|Q, \mathcal{G})$ as shown in Eqs. (13,14,15).

### 3.6 OVERVIEW AND IMPLEMENTATION

Before verifying the effectiveness of our `CoD`, we give detailed implementations to ensure the reproducibility of our experiments. We first fine-tune a small generative model $\phi_{\theta_e}$, i.e., T5, to retrieve question-relevant context. Then, we reformulate the retrieved context using a hand-crafted rule, which is parameter-free and achieved by the function $h_\emptyset$. Finally, we fine-tune LLMs $f_{\theta_r}$ to generate a logical form and identify whether the reasoning path can derive correct answers. Namely, for a given input query $Q = q$ and a structured knowledge base $\mathcal{G}$, our method `CoD` predicts by

$$\hat{\mathcal{A}}, \hat{\mathcal{M}} = \underbrace{f_{\theta_r}}_{\text{reason}} \circ \underbrace{h_\emptyset}_{\text{reformulate}} \circ \underbrace{\phi_{\theta_e}(q, \mathcal{G})}_{\text{retrieve}}, \tag{16}$$

where $\hat{\mathcal{A}}$ and $\hat{\mathcal{M}}$ denote the set of the predicted answers and LFs. This is because `CoD` generates a pair of answers and LFs for each reasoning path. In this regard, we follow previous work to predict answers using executable LFs with the highest confidence. If the predicted LFs $\hat{M}$ is non-executable, the predicted answer is constructed using all tail entities of reasoning paths with $a = $ "yes".

## 4 RELATED WORK

**Knowledge retrieval in KBQA.** In multi-hop KBQA scenarios, the goal of knowledge retrieval is to extract relevant documents from the knowledge graph that are relevant to the given question. The process begins with initial topic entities, from which it is necessary to select relevant neighboring triples from large-scale KGs to form a path leading to the answer entities. Traditional lexical models such as BM25 (Robertson et al., 2009) and PageRank (Sun et al., 2018; He et al., 2021) have been extensively employed in KBQA. However, these methods often overlook the semantics of the question, potentially impairing the efficiency and accuracy of retrieval. Recent advancements in the field have seen the emergence of dense semantic retrieval models, including Dense Passage Retrieval(Karpukhin et al., 2020), SimCSE (Gao et al., 2021), and Contriever (Izacard et al., 2021), employ bi-encoder architectures to transform sentences into low-dimensional dense vectors.

**Knowledge reasoning in KBQA.** The reasoning stage in KBQA focuses on accurately identifying answer entities by traversing relations starting from the topic entities. Early approaches (Miller

---

[4]While these independence assumptions provide theoretical connections to prior work, they serve as analytical tools rather than operational requirements; the effectiveness of CoD is ultimately validated empirically.

et al., 2016; Sun et al., 2018; 2019; Jiang et al., 2022) employ specialized network architectures such as Key-Value Memory Networks to model the multi-hop reasoning process. Recent works have proposed parsing questions into structured query languages (e.g., logical forms (LFs)) and executing them via query engines to obtain answers (Lan et al., 2019; Das et al., 2021; Huang et al., 2021). These methods typically use encoder-decoder architectures, such as T5 (Raffel et al., 2020), to generate structured queries. In the area of LLMs, DECAF (Yu et al., 2022) combines LF and LLMs reasoning to jointly generate answers. ChatKBQA (Luo et al., 2023a) generates logical forms through fine-tuning LLMs based solely on the input question, and obtains the final answers through query engines. The generated logical forms may be inaccurate. To effectively utilize the path information of knowledge graphs, some studies use subgraphs obtained through retrieval for reasoning. TOG (Sun et al., 2023) leverages the reasoning capabilities of open-source LLM, iteratively exploring various possible reasoning paths on the KG through prompt tuning, until the LLM determines that it can answer the question based on the current reasoning path. ROG (Luo et al., 2023b) proposes a planning retrieval reasoning framework that synergizes LLM to achieve faithful and explainable reasoning. More details can be found in Appendix E.

## 5 EXPERIMENTS

### 5.1 SETUP

**Datasets.** We conduct experiments on two benchmark KBQA datasets: **WebQuestionSP (WebQSP)** (Yih et al., 2016) and **ComplexWebQuestions (CWQ)** (Talmor & Berant, 2018), both available for Freebase KB reasoning. The former contains 4,737 simple natural language questions with SPARQL queries, while the latter contains 34,689 more complex questions with SPARQL queries. Training and testing splits of each dataset follow the same setting as those of previous work (Yu et al., 2022).

Table 2: Performance comparison of different retrieval models on the two KBQA datasets.

| Method | WebQSP | | | CWQ | | |
|---|---|---|---|---|---|---|
| | top5-acc | top10-acc | top20-acc | top5-acc | top10-acc | top20-acc |
| DPR (Karpukhin et al., 2020) | 35.02 | 44.17 | 53.87 | 10.24 | 17.90 | 28.14 |
| BM25 (Robertson et al., 2009) | 15.25 | 20.50 | 22.70 | 2.77 | 6.59 | 11.95 |
| Sentence-BERT | 30.69 | 43.32 | 55.83 | 9.13 | 17.10 | 32.82 |
| CoD (ours) | **91.04** | **95.00** | **96.77** | **89.93** | **92.82** | **95.11** |

**Evaluation Metrics.** To obtain a fair comparison, we adopt the F1 score and Hits@1 to evaluate the coverage of all answers and the single top-ranked answer, respectively, following previous work (Shu et al., 2022; Yu et al., 2022). In terms of retriever comparison, top-k accuracy is utilized to demonstrate whether the correct reasoning paths are searched within the top-k retrieval results.

**Baselines.** The baselines include: Non-LLMs-based method, Prompting-LLM-based method, and Finetuning-LLM-based method. Below are the details. (1) Non-LLMs based method. TIARA (Shu et al., 2022) improves question answering by using multi-grained retrieval and constrained decoding to enhance the accuracy and robustness of KBQA. UniKGQA (Jiang et al., 2023) unifies the retrieval and reasoning in both model architecture and parameter learning. HGNet (Chen et al., 2022) proposes a hierarchical query graph generation approach to improve the performance of KBQA. (2) Prompting-LLM-based method. ToG (Sun et al., 2023) involves prompting the LLM to iteratively explore various potential reasoning paths within KGs until it concludes that the question can be answered based on the retrieved path. FiDeLis (Sui et al., 2024) proposes a retrieval-exploration method that incorporates knowledge graphs into LLMs, exploiting their deductive reasoning abilities to enhance knowledge retrieval and reasoning. (3) Finetuning-LLM based method. KD-CoT (Wang et al., 2023a) extracts relevant knowledge from KGs to create accurate reasoning paths for LLMs. DeCAF (Yu et al., 2022) integrates semantic parsing with LLM reasoning to jointly produce answers, achieving notable performance in KGQA tasks. RoG (Luo et al., 2023b) incorporates structural knowledge from knowledge graphs (KGs) into neural networks during pretraining and fine-tuning. We use Llama2-7B as the base LLM, with a batch size of $4$ and a learning rate of $5e^{-5}$. More details can be found in Appendix F.

Table 3: Performance evaluation of the proposed `CoD` with different hyperparameters, i.e., beam sizes.

| Beam Size | F1% on WQSP | | | F1% on CWQ | | |
|---|---|---|---|---|---|---|
| | Only LF | Only RP | CoD | Only LF | Only RP | CoD |
| bm=1 | 71.60 | 74.21 | 75.56 | 44.55 | 45.65 | 62.13 |
| bm=2 | 74.46 | 76.67 | 78.37 | 47.48 | 50.09 | 65.83 |
| bm=3 | 75.04 | 77.26 | 80.87 | 48.25 | 50.67 | **68.27** |
| bm=5 | **79.60** | **78.62** | **81.24** | **49.40** | **51.37** | 65.70 |

## 5.2 MAIN RESULTS

Table 1 illustrates the comparative performance of our method against various baseline methods. `CoD` improves Hits@1 and F1 by over 2% and 2% on the WebQSP dataset and by over 5% and 1% on the CWQ dataset, respectively, compared to the previous optimal methods. The data clearly indicate that our method surpasses all baselines, even those enhanced with GPT-4-turbo, including strong contenders, like DeCAF and FiDeLis. Among all methods based on prompting, GPT-4-turbo consistently delivers superior results compared to GPT-3.5-turbo, particularly on the CWQ dataset, suggesting its enhanced ability to comprehend and process complex queries.

Among them, the non-LMs-based methods have limited performance improvements due to their limited reasoning capabilities. The Prompting-LLM Only-based methods have poor performance due to the lack of knowledge graphs. For the methods based on Prompting-LLM + KG and Finetuning-LLM + KG, RoG combines LLM with KG to achieve faithful and explainable reasoning. FiDeLis uses the retrieval-exploration-interactive framework to enable large models to perform KBQA in a deductively verified manner, making full use of the advantages of LLM in logical reasoning, and it achieves state-of-the-art performance. However, this kind of method may introduce too many paths, resulting in limited performance improvement. Our method effectively improves the accuracy of knowledge graph retrieval by designing an enhanced retriever.

## 5.3 RETRIEVERS PERFORMANCE COMPARISON

To verify the effectiveness of the proposed retriever, we conducted the experiments as shown in Table 2. It can be seen that our method is significantly higher than DPR, BM25, and Sentence-BERT. In other words, when performing knowledge reasoning, our method only needs to retrieve a small number of paths to fully include the required reasoning paths, effectively reducing the impact of unnecessary paths on the performance of the model in the reasoning stage. Additionally, we paired CoD with different retrievers. The corresponding results and analysis are presented in Appendix D.

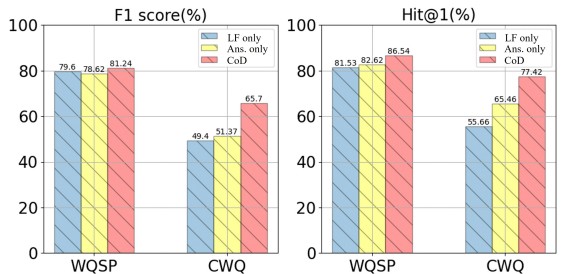

Figure 3: The performance of `CoD` with only generated answers (indicated as Ans. only) or LF executed answers (indicated as LF only).

## 5.4 ABLATION STUDIES

Beam size decides the maximum number of candidate outputs remaining after beam search. We conduct experiments to verify its impact on model performance. As shown in Table 3, with the increase in beam size, the performance of the model gradually improves, and when bm=3, the performance gradually stabilizes. This suggests that, when generating answers through beam search, LLM tends to provide the correct answer within the top-generated responses. To verify the significance of the combination of LF-executed answers and generated answers in `CoD`, we conducted ablation studies with results in Figure 3 when bm=5, where the performance of joint utilization surpasses that of either the answer-only or LF-only approaches. The effectiveness is more distinct for the CWQ dataset, indicating robustness when encountering complicated KB questions. Additionally,

experiments on the impact of different prompt templates and robustness analysis are presented in Appendix D. To further evaluate generalization, we test CoD on GrailQA across i.i.d., compositional, and zero-shot settings. CoD achieves 78.2% overall F1, outperforming DECAF (76.0%) by 2.2 percentage points, with consistent gains in compositional (+0.9%) and zero-shot (+1.3%) splits, demonstrating that the decomposition strategy transfers robustly across different distribution shifts.

## 6  CONCLUSION

In this paper, we present Chain-of-Decomposition (CoD), a novel framework that addresses the challenges of adapting large language models (LLMs) to Knowledge Base Question Answering (KBQA). Based on our causal graph, we factorize the answer-generation process and isolate computation-heavy retrieval and rule-based reformulation from LLM reasoning. We decompose KBQA into three modular steps—an LLM-free retrieval module, a parameter-free reformulation step, and a lightweight LLM-based reasoning module— significantly reduce task complexity and enable efficient fine-tuning of LLMs. Experiments on benchmarks demonstrate that Llama-2 7B fine-tuned with CoD outperforms strong baselines, including GPT-4 augmented with retrieved knowledge, achieving state-of-the-art performance. The efficacy of the proposed CoD highlights the importance of task decomposition in enhancing LLM reasoning capabilities for complex reasoning tasks.

**Limitation.** While CoD achieves notable advancements, it has limitations. We solely employ Llama-2 7B as the base LLM, overlooking more advanced models like Llama-3 (Grattafiori et al., 2024), QWen, and DeepSeek-R1 (Guo et al., 2025). Exploring these advanced LLMs to further improve CoD's performance is a promising direction for future work.

## ACKNOWLEDGMENTS

We thank the reviewers and area chairs for their valuable and constructive comments, which have significantly improved the quality of this paper.

## REPRODUCIBILITY STATEMENT

We outline the pipeline of our proposed method in Fig. 1 and provide implementation details in Appendix B. Key data and settings are also included in Appendix A for reference.

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

APPENDIX

## A    SETTING OF KNOWLEDGE BASE QUESTION ANSWERING

Knowledge base question answering (KBQA) is a complex task that requires *factual knowledge* and *deep reasoning* to accomplish accurately. In KBQA, factual knowledge typically is in the form of a set of triples, i.e., *subject entity*, *object entity*, and the *relation* of these two entities. The deep reasoning refers to the fact that the input query typically requires complicated multi-hop reasoning. Namely, on the knowledge graph $\mathcal{G}$, the answer entities are multiple hops away from the topic entities mentioned in the natural language query.

There are mainly two datasets used in the literature, i.e., WebQuestionSP (WebQSP) (Yih et al., 2016) and Complex WebQuestions (CWQ) (Talmor & Berant, 2018). These two datasets result from the Freebase (Bollacker, 2008) that contains around 88 million entities, 20 thousand relations, and 126 million triples. The data split strategy is the same as previous work (Sun et al., 2018; Luo et al., 2023b). Specifically, $2,826$ samples are used for training, while $1,628$ is used for testing on the WebQSP dataset. For CWQ, we leverage $27,639$ samples for model training and $3,531$ for testing. Here, the sample comprises a pair of the input query and the corresponding answers[5].

These two datasets contain up to 4-hop questions, which require deep reasoning. The deep reasoning is challenging even for large language models (LLMs). For instance,

---

**Query:**
Which book written by the author who married the actress starring in the movie directed by the director who won an Oscar for a movie starring Tom Hanks?

**Multi-hop reasoning:**

Tom Hanks $\xrightarrow{\text{starred\_in}}$ Forrest Gump $\xrightarrow{\text{directed\_by}}$ Robert Zemeckis $\xrightarrow{\text{married\_to}}$

Richard Matheson $\xrightarrow{\text{author\_of}}$ I Am Legend

---

We can see that a 4-hop question is challenging even for humans.

## B    TEMPLATE DESIGN OF REASONING PATH

According to our factorization, we show that the reformulating model could be handcrafted rules, leading to a parameter-free algorithm. Namely, we can design a template to convert the retrieved context into the required reasoning paths. Here, we provide details for the template design.

For the retrieved context, i.e., a set of triples, we group triples into four classes according to the number of hops. Then, we determine a series of candidate reasoning paths based on the principle of whether the head entity in the $i$-th hop is consistent with the tail entity in the $i + 1$-th hop. The overview of the reformulating model can be found in Algorithm 1.

---

**Algorithm 1** Extract Reasoning Path for Input Query

---

1: **procedure** EXTRACT REASON PATH($data$)
2:     $t \leftarrow data[\text{“triplets”}]$
3:     # Link the triplets
4:     $linked\_t \leftarrow$ LINK_TRIPLETS($t$)
5:     # Generate reasoning paths
6:     $reason\_p \leftarrow$ GENERATE_PATH($linked\_t$)
7:     # Replace entity indexes with entity names
8:     $new\_path \leftarrow$ GENERATE_WITH_NAME($reason\_p$)
9:     **return** $new\_path$
10: **end procedure**

---

[5]In these two datasets, the SPARQL query is also provided. Thus, we can convert the SPARQL of each input query into logical forms.

In our experiments, the retrieval model can filter out most irrelevant relations. Namely, we have a limited number of tuples in each group with the same hop. Moreover, the number of tuples that link head-to-tail entities between different groups is also usually limited. Thus, the reformulating model typically results in a limited number of reasoning paths, i.e., less than $10$.

## C  MATHEMATICAL CONNECTION TO ROG

We construct a general causal graph by rethinking the widely adopted approaches to knowledge base question answering (KBQA). Namely, the causal graph formulates the generation process of answer $A$ or logical form $M$ for a given input query $Q$. Our factorization approach results from the general causal graph. Thus, our factorization is a general approach that could be realized using different assumptions.

We show that, under some assumptions, a recent outstanding approach, RoG (Luo et al., 2023b), for KBQA is a special case of our factorization. To clarify the connection, we first list two assumptions used in RoG.

**Assumption 1.** *When conditioning on the context $Z$, the generation of the answer $A$ to the query $Q$ is independent of the question-relevant context $C$.*

This assumption is intuitive, as the query and reasoning path can determine the answer.

**Assumption 2.** *The generation of reasoning paths $Z$ is independent of the knowledge base $\mathcal{G}$ conditioning on the input query $Q$.*

However, this assumption could be violated in many practical scenarios. In particular, the reasoning path is based on the knowledge graph (or knowledge base) $\mathcal{G}$. Namely, the reasoning path usually varies with the utilized knowledge graph $\mathcal{G}$. Thus, the assumption could be relatively strong for many real-world scenarios.

Based on these two assumptions, we can show that our factorization $\sum_Z P(A|Q,Z) \sum_C P(Z|C)P(C|Q,\mathcal{G})$ gives a general formulation and the objective used in RoG $\sum_Z P(A|Q,\mathcal{G},Z)P(Z|Q)$ is a special case of our factorization.

$$\sum_Z P(A|Q,Z) \sum_C P(Z|C)P(C|Q,\mathcal{G})$$

$$=P(A|Q,\mathcal{G}) \tag{17}$$

$$=\sum_Z \sum_C P(A,Z,C|Q,\mathcal{G}) \tag{18}$$

$$=\sum_Z \sum_C P(A|Q,\mathcal{G},Z,C)P(Z,C|Q,\mathcal{G}) \tag{19}$$

$$=\sum_Z \sum_C P(A|Q,\mathcal{G},Z,C)P(C|Q,\mathcal{G},Z)P(Z|Q,\mathcal{G}) \tag{20}$$

$$=\sum_Z \sum_C P(A|Q,\mathcal{G},Z)P(C|Q,\mathcal{G},Z)P(Z|Q,\mathcal{G}) \tag{21}$$

$$=\sum_Z \sum_C P(A|Q,\mathcal{G},Z)P(C|Q,\mathcal{G},Z)P(Z|Q) \tag{22}$$

$$=\sum_Z \left[ \sum_C P(C|Q,\mathcal{G},Z) \right] P(A|Q,\mathcal{G},Z)P(Z|Q) \tag{23}$$

$$=\sum_Z P(A|Q,\mathcal{G},Z)P(Z|Q), \tag{24}$$

where Eq. (21) holds based on Assumption 1, Eq. (22) holds based on the Assumption 2, and Eq. (24) is exactly the learning objective of RoG (Luo et al., 2023b). Thus, RoG is a special case of our factorization under Assumptions 1 and 2.

## D  ADDITIONAL EXPERIMENTS

In this section, we further present the experimental results and in-depth analysis for additional research questions.

**Robustness analysis.** To evaluate the robustness of the proposed CoD method, we conduct experiments using different backbone LLMs. As shown in Table 4, we extend our evaluation to two additional base LLMs: Llama3-8B and Qwen2.5-7B. These models are fine-tuned within the CoD framework and evaluated on two knowledge-based question answering (KBQA) benchmarks, namely WebQuestionsSP (WQSP) and ComplexWebQuestions (CWQ). The results indicate that CoD maintains robust performance across different LLMs, underscoring its adaptability and generalizability.

Table 4: The performance of CoD across different backbone LLMs on the WQSP and CWQ datasets.

| Model | WQSP | | CWQ | |
|---|---|---|---|---|
| | F1 | Hits@1 | F1 | Hits@1 |
| Llama2 7B | 81.24 | 86.54 | 65.7 | 77.42 |
| Llama3 8B | 81.03 | 86.29 | 79.61 | 79.33 |
| Qwen2.5 7B | 80.39 | 85.42 | 66.53 | 77.85 |

Table 5: The impact of various retrievers on the performance of the CoD model on the WQSP and CWQ datasets.

| Model | WQSP | | CWQ | |
|---|---|---|---|---|
| | F1 | Hits@1 | F1 | Hits@1 |
| CoD (DPR) | 78.71 | 83.37 | 60.13 | 73.15 |
| CoD (BM25) | 77.93 | 81.33 | 59.52 | 71.83 |
| CoD (SentenceBERT) | 83.52 | 60.62 | 66.53 | 74.02 |
| CoD (Ours) | 80.39 | 86.54 | 65.70 | 77.42 |

Table 6: The impact of different prompt templates on the performance of the CoD model.

| Model | WQSP | | CWQ | |
|---|---|---|---|---|
| | F1 | Hits@1 | F1 | Hits@1 |
| CoD (Direct Generaton) | 77.53 | 82.52 | 62.77 | 75.16 |
| CoD (CoT) | 79.36 | 86.15 | 64.94 | 76.39 |
| CoD (Ours) | 81.24 | 86.54 | 65.70 | 77.42 |

**The impact of different retrievers.** To investigate the impact of different retrievers on the performance of the CoD model, we conduct experiments as shown in Table 5. The results demonstrate that CoD achieves promising performance when using our specially designed retriever. This finding underscores the importance of accurate retrieval paths for the effectiveness of our approach. The comparative analysis with other popular retrievers (DPR, BM25, and Sentence-BERT) further validates the necessity of our retrieval component design.

**The impact of different prompt templates.** To investigate the impact of different prompt templates on the CoD model's performance, we conduct experiments as shown in Table 6, we evaluate three answer generation templates using the retrieved reasoning paths: direct generation, Chain-of-Thought (CoT), and our proposed deductive reasoning approach. The choice of prompt template indeed has a impact on the performance of the CoD method. Both Chain-of-Thought (CoT) and our deductive reasoning approach outperform direct generation. Furthermore, our method demonstrates slightly superior performance to CoT and exhibits better robustness and stability across different templates.

# E  ADDITIONAL RELATED WROK

To effectively utilize the path information of knowledge graphs, some studies use subgraphs obtained through retrieval for reasoning. PullNet (Sun et al., 2019) retrieves a relevant subgraph from the KB and uses graph neural networks to predict answer entities within these subgraphs. UniKGQA (Oguz et al., 2020) unifies the knowledge base retrieval and reasoning processes into a single model with PLM, achieving SOTA performance. (Li et al., 2023) break down questions into subquestions and generate SPARQL queries using a fine-tuned llama model (Touvron et al., 2023) to fetch knowledge from KGs. (Wu et al., 2023) make full use of the retrieved knowledge of the knowledge graph through steps such as Retrieve-Rewrite-Answer and fine-tuning the large model to obtain the final answer.

# F  IMPLEMENTATION DETAILS

All experiments are conducted on a single NVIDIA A40 GPU. For the retrieval module, we fine-tune a T5-base model using the AdamW optimizer ($\beta_1$=0.9, $\beta_2$=0.999) with a learning rate of 2e-5 and a batch size of 4, training for 100 epochs on both WebQSP and CWQ, which takes approximately 3 hours and 8 hours respectively. For the reasoning module, we fine-tune LLaMA-2-7B using LoRA (rank=8, $\alpha$=16) with a learning rate of 5e-5, a batch size of 4, and a maximum sequence length of 2048. The reasoning module is trained for 100 epochs on WebQSP ( 12 hours) and 5 epochs on CWQ ( 24 hours).

