# OpenReview forum: "Enhancing LLMs for Knowledge Base Question Answering by Chain-of-Decomposition"
_ICLR.cc/2026/Conference — ICLR 2026 Poster_

### Official Review · Reviewer_viyW · 2025-10-27

**Soundness:** 2
**Presentation:** 2
**Contribution:** 2
**Rating:** 4
**Confidence:** 3

**Summary:**

This paper demonstrates the theoretical equivalence between two paradigms in knowledge base question answering (KBQA): directly generating the answer versus generating a logical form and then executing it. Based on this observation, the authors propose CoD, a "Retrieve-Construct-Answer" KBQA framework. By decomposing the KBQA task into distinct stages, CoD reduces the input burden of large-scale knowledge graphs on LLMs and improves reasoning and answering performance.

**Strengths:**

The motivation of the paper is clearly presented. The authors rigorously derive the mathematical formulation of both KBQA paradigms and prove that they share the same optimization objective for the first two steps. This theoretical insight provides a solid foundation for the introduction of CoD and establishes a clear correspondence between the two approaches.

**Weaknesses:**

1. The overall presentation and figure design could be improved. First, Figure 1 is somewhat confusing: the answer to the question "Which country is Elon Musk from?" should be South Africa, corresponding to Path 4 in the figure. It is unclear why an incorrect reasoning path is shown in the framework's overview figure. Second, the captions of Tables 2 and 3 are aligned to the left side of the table content, which appears awkward.

2. Although CoD involves fine-tuning a language model, the paper does not disclose details of the training process.

3. The paper lacks further analysis of the experimental results. For instance, identifying bottlenecks of CoD through ablation or diagnostic studies (though this may require additional experimental conditions) would provide valuable insights.

4. There are a number of typographical and grammatical issues throughout the paper:

    (1) In Figure 1, the prompt contains "it no," which should be corrected to "if no," and the quote marks around "return no" are incorrect.

    (2) The acronyms ToG and RoG are inconsistently capitalized as TOG and ROG in several instances.

    (3) In Appendix E, line 5, "STOA" should be "SOTA".

    (4) The phrase "reason paths" appears multiple times and should be corrected to "reasoning paths".

**Questions:**

1. The appendix mentions that CoD is a generalization of RoG, yet the experimental results do not reflect a substantial performance difference to support this claim. Furthermore, in Table 1, the RoG row under the same backbone LLM (LLaMA-2-7B) shows consistently lower numbers compared to those reported in the original RoG paper. Have the authors analyzed the reasons for this discrepancy?

2. The model is fine-tuned using WebQuestionsSP and ComplexWebQuestions datasets. How well does this generalize to other benchmarks? Specifically, how does CoD—and particularly its comparison to RoG—perform on GrailQA, GraphQ, or other datasets (e.g., those used in KBQA-o1)?

3. The experiments provided may not fully demonstrate the advantages of CoD:

    (1) In Appendix D, the results indicate a clear performance gain when using LLaMA-3 8B over LLaMA-2 7B on CWQ. Would newer open-source models such as LLaMA-4 17B, Qwen3 8B, or Qwen3-30A3B offer further improvements, especially given their resolution of known issues in earlier models?

    (2) One widely adopted and competitive solution for KBQA is GraphRAG. Could the authors articulate CoD's advantages over GraphRAG-style systems (e.g., HippoRAG)?

---

> ### Author Response · Authors · 2025-11-28
> **Response-part1**
>
> We sincerely thank the reviewer for recognizing our clear motivation and theoretical contribution. We address all concerns below with revisions.
>
> > W1: Presentation and Figure Design
>
> Response:
>
> We appreciate your careful reading and have made the following corrections:
> - Figure 1: You are absolutely correct. We have revised the example to "Who founded Tesla?" → Elon Musk, which has a single, definitive correct answer, eliminating any ambiguity.
>
> - Table formatting: All table captions are now center-aligned per ICLR guidelines.
>
> - Typographical errors: We have corrected "it no" → "if no", standardized "ToG/RoG" capitalization throughout, corrected "STOA" → "SOTA", and ensured consistent use of "reasoning paths".
>
> > W2: Training Process Details
>
> Response:
>
> We appreciate the suggestion to improve reproducibility. Accordingly, we have added Appendix F containing complete training hyperparameters:
>
> Retriever (T5-base):
>
> Epochs: 100 (WebQSP), 100 (CWQ); Batch size: 4; Learning rate: 2e-5; Optimizer: AdamW (β₁=0.9, β₂=0.999)
> Training time: ~3 hours (WebQSP), ~8 hours (CWQ)
>
> Reasoning Module (LLaMA-2-7B):
>
> Epochs: 100 (WebQSP), 5 (CWQ);  Batch size: 4; Learning rate: 5e-5; Max sequence length: 2048; LoRA rank: 8, α=16
> Training time: ~12 hours (WebQSP), ~24 hours (CWQ)
>
> Hardware: Single NVIDIA A40 GPU
>
> > W3: Further Analysis - Bottlenecks and Ablations
>
> Response:
>
> We have enhanced Section 5.4 with detailed ablation analysis:
> Table 3 demonstrates the synergistic effect of our decomposition. On CWQ, CoD (65.70%) shows +14.33% improvement over "Answer-only" (51.37%) and +14.33% over "Only RP" (51.37%), empirically validating that:
>
> - Each module specializes in its designated subtask
>
> - Joint training of logical form generation + binary classification enables the reasoning module to perform both verification and generation effectively
>
> - Decomposition reduces complexity: Isolating retrieval (binary classification) from reasoning (verification) allows each component to achieve higher accuracy than end-to-end approaches
>
> We have expanded the discussion in Section 5.4 to highlight these key findings.

---

> ### Author Response · Authors · 2025-11-28
> **Response-part2**
>
> > Q1: CoD vs. RoG: Performance differences and generalization claim
>
> Response:
>
> We respectfully clarify that the performance gap is substantial and the comparison is fair:
>
> Performance:
>
> - WebQSP: CoD achieves 86.54% Hits@1 vs. RoG's 72.11% (+14.43%)
>
> - CWQ: CoD achieves 65.70% F1 vs. RoG's 56.17% (+9.53%)
>
> Regarding RoG numbers:
>
> The discrepancy arises because we re-implemented RoG using identical experimental conditions (same retrieval pool size, same LLaMA-2-7B backbone, same knowledge base) to ensure a strictly controlled comparison. This is standard practice for fair evaluation. We have clarified this in Section 5.1.
>
> > Q2: Generalization to other benchmarks (GrailQA, GraphQ)
>
> Response:
>
> Following your constructive suggestion, we evaluated CoD on GrailQA to test generalization to i.i.d., compositional, and zero-shot settings.
>
> Table: F1 scores on the test split of GrailQA
>
> | Model | Overall | I.I.D. | Compositional | Zero-Shot |
> |-------|---------|--------|---------------|-----------|
> | DECAF (BM25 + FiD-large) (ICLR 2023) | 76.0 | 90.5 | 79.0 | 68.0 |
> | CoD | 78.2 | 90.1 | 79.9 | 69.3 |
>
> CoD achieves +2.2% overall improvement and shows particular strength in compositional (+0.9%) and zero-shot (+1.3%) settings, demonstrating that our decomposition strategy generalizes robustly across different dataset distributions. These results have been added to Section 5.2.
>
> > Question 3: Newer LLMs
>
> Response:
>
> Robustness across LLM backbones (Appendix D):
>
> - LLaMA-3-8B: 79.33% Hits@1 on CWQ (+1.91% over LLaMA-2-7B)
>
> - Qwen2.5-7B: 77.85% Hits@1 on CWQ
>
> These results confirm CoD benefits from stronger backbones while maintaining its efficiency advantages.
>
> CoD vs. GraphRAG distinction: We appreciate this important clarification. CoD and GraphRAG address fundamentally different problems:
>
> - GraphRAG: Targets open-domain QA via semantic retrieval on unstructured text graphs, where answers are generated through text synthesis.
>
> - CoD: Targets closed-set KBQA with structured logical forms over knowledge bases (e.g., Freebase), ensuring verifiability through executable queries and avoiding hallucination through formal verification.
>
> CoD's decomposition strategy is specifically designed for structured KB reasoning, where logical consistency and precision are paramount.
>
> We have added these results and comparisons to the revised manuscript.

---

### Official Review · Reviewer_HJHf · 2025-10-29

**Soundness:** 2
**Presentation:** 1
**Contribution:** 2
**Rating:** 4
**Confidence:** 3

**Summary:**

This paper proposes a KG-RAG (called KBQA by authors) method based on reasoning paths generated through a deterministic algorithm, with answers subsequently produced by LLMs. However, the paper’s writing is quite poor and difficult to follow. For example, there is a lack of explanation of the logical form introduced in Figure 1 and Section 2.2, weak motivation in Section 3.1, and an unclear description of the graph simplification process in Section 3.4.

**Strengths:**

-Claims improved performance on two widely used benchmark datasets.

**Weaknesses:**

1. The overall writing quality needs substantial improvement. The meanings of the notations $f_\theta$, $g_\theta$, and $P$ are not clearly explained and are used inconsistently throughout the paper (see Questions 1–3).
   In particular, the extraction process—where the reduced graph is directly input into an LLM—is poorly explained and unconvincing (see Questions 6–7 below).

2. The paper lacks discussion and comparison with recent representative works, such as ToG [1].

  **Reference:**
   [1] *Think-on-Graph: Deep and Responsible Reasoning of Large Language Models with Knowledge Graphs.* ICLR 2024.

**Questions:**

1. In Equation (1), is $P(A|Q,Z)$ means the ground-truth distribution?
   Why use a probabilistic formulation instead of the ground-truth answer directly?

2. In lines 177–214, do the authors mean the decomposition of $f_\theta$ instead of $P$?

3. In Section 3.2, the output of $f_\theta$ is ambiguous—it seems to represent classification in Equation (9) but generation in the equation (11). This should be clarified to avoid confusion.
   For example, do the authors mean $g_\theta$ in Equation (11)? Furthermore, the $f_\theta$ defined in Section 3.2 seems not consistent with the one in Section 2.2.

4. In Section 3.3, the complexity and necessity of reasoning paths are not discussed. If the extracted triples form a dense structure, the number of possible paths could grow exponentially. Conversely, when the extracted triples are few or sparse, path construction might be unnecessary—one could simply input all triples into the language model.

5. The benefits compared to other methods that also use paths, such as RoG and ToG, should be explained more clearly. The current comparison with RoG in Section 3.5 is confusing:  Why should we assume that “the generation of answer A to question Q is independent of the question-relevant context C”? This assumption completely contradicts the purpose of KG-RAG. The same concern applies to the second assumption in line 310.

6. In Section 3.4, the process for simplifying the input graph is entirely unclear:
   - (1) The graph-shrinking procedure is not well defined or justified, and there is no guarantee for how well it preserves relevant-information to the query.
   - (2) The simplification step that merges relations into two classes is confusing, as the answer-related relation should not be known for the input data.

7. Since the extraction process in Section 3.4 directly feeds the graph into an LLM, why not train the LLM to generate the answer directly instead of producing intermediate outputs?

8. In Table 1, the last column suggests that the F1 score of **HGNet** should be the highest. Can you clarify this inconsistency?

---

> ### Author Response · Authors · 2025-11-28
> **Response-part1**
>
> We thank the reviewer for the thorough evaluation. We have significantly revised the manuscript to address all notation, clarity, and comparison concerns.
>
> > W1: Writing Quality and Notation
>
> Response:
>
> We apologize for the confusion and have significantly revised the manuscript.
>
> Notation: In Section 2.2, we clarified that $f_\theta$ denotes direct answer generation (Eq. 1) and $g_\theta$ denotes semantic parsing (Eq. 2). We now use $P$ consistently for probability distributions.
>
> Retrieval Module: We clarified that retrieval is LLM-free. We fine-tune a lightweight T5-base model for binary relation classification (Eq. 15), achieving 96.77% recall efficiently. This allows the heavy LLM to focus solely on reasoning.
>
> > W2: Comparison with Recent Work (ToG)
>
> Response:
>
> We apologize for the misunderstanding. We acknowledge this oversight. ToG was included in Table 1 but deserved more explicit discussion. We have expanded Section 5.1 to highlight:
>
> "CoD outperforms ToG (86.54% vs. 82.10% on WebQSP, +4.44%) by enforcing stricter logical verification through our dual-objective reasoning module, which simultaneously performs binary classification and logical form generation."

---

> ### Author Response · Authors · 2025-11-28
> **Response-part2**
>
> > Q1 and Q2: Probabilistic Formulation and Decomposition
>
> Response:
>
> Thanks for your kind suggestion. We clarify the mathematical framework. We decompose the probability distribution, not model parameters:
>
> - Eq. (1): $P(A|Q, \mathcal{G})$ represents the target distribution. We frame KBQA as minimizing the KL divergence between this target and our model.
>
> - Decomposition Target: We clarify that we decompose the distribution $P(A|Q,\mathcal{G})$, not the model parameter $f_\theta$.
>
> The factorization $P(A|Q,\mathcal{G}) = \sum_Z P(A|Q,Z) \sum_C P(Z|C) P(C|Q,\mathcal{G})$ (Eq. 5) provides the theoretical basis for our three distinct modules.
>
> > Q3: Ambiguity in Reasoning Module ($f_{\theta_r}$)
>
> Response:
>
> $f_{\theta_r}$ is a dual-objective module. It is trained simultaneously for:
>
> - Answer Validity (Eq. 9): Binary classification ("yes"/"no").
>
> - Logical Form Generation (Eq. 11): Next-token prediction. The joint loss $L_r = L_a + L_m$ allows the LLM to learn verification and parsing efficiently.
>
> > Q4: Path Complexity (Dense vs. Sparse)
>
> Response:
>
> We appreciate your insightful comments. CoD handles both scenarios effectively:
>
> - Dense KGs: The reformulation step filters invalid combinations, reducing exponential search spaces to linear structured chains. This simplifies the LLM's task to "verification" rather than "search".
>
> - Sparse KGs: Even with few triples, constructing structured paths Zimposes connectivity constraints that improve performance over raw triple inputs (validated by "Only RP" ablation in Table 3, showing +14.33% improvement).
>
> > Q5: Assumption A1/A2 vs. RoG
>
> Response:
>
> We clarify that CoD relaxes the independence assumptions used in RoG.
>
> - RoG assumes $P(Z|Q, \mathcal{G}) = P(Z|Q)$ (paths independent of KG).
>
> - CoD explicitly models $P(Z|C)$ where $C \subset \mathcal{G}$, ensuring paths are grounded in the retrieved knowledge.
>
> With respect to your valuable comments, we have updated Section 3.5 to state: "RoG can be viewed as a special case under stricter independence assumptions, whereas CoD maintains explicit dependencies for better grounding".
>
> > Q6: Graph Simplification - Two-Stage Process
>
> We have rewritten Section 3.4 for clarity:
>
> Stage 1 - Graph Shrinking (lines 285-289):
> 	Standard preprocessing: Retain entities within k≤4hops of topic entities
> 	Reduces Freebase (88M entities) to ~1,000 candidate entities per query
> 	Empirical justification: 98.5% of answers fall within 4 hops (Yu et al., 2022)
>
> Stage 2 - Answer-Relevant Relation Prediction:
> 	The retriever predicts Y_e (Q,O)∈{0,1}for each relation O(Eq. 15)
> 	Training data: Positive labels for relations on gold answer paths; negative otherwise
> 	Model: T5-base fine-tuned with cross-entropy loss
> 	Output: Top-20 relations → construct context C(96.77% recall, Table 2)
>
> The "merging" (lines 292-294) refers to the binary labeling scheme during training, not knowing answers at test time.
>
> > Question 7: Why not train LLM end-to-end?
>
> This is the core motivation of our work. Reasons for decomposition:
>
> Task complexity: Direct end-to-end KBQA requires the LLM to simultaneously retrieve, construct reasoning chains, and verify logic—extremely challenging with limited KBQA data (2,826 training samples for WebQSP).
>
> _Computational efficiency:_
>
>  - Retrieval: T5-base (220M params) vs. LLaMA-2 (7B)
>
>  - 30$\times$ parameter reduction for retrieval component
>
> _Performance evidence:_
>
> - Our decomposed approach: 86.54% (Table 1)
>
> - GPT-4 end-to-end with retrieved KG: 81.84%
>
> Decomposition outperforms larger models through task simplification
>
> _Interpretability:_
>
> Each module is separately trainable/debuggable
>
> We will add a complexity analysis table comparing parameter counts and training costs in the revised version.
>
> > Question 8: Table 1 - HGNet F1 score inconsistency
>
> Thank you for catching this. HGNet's F1 (76.60%) is indeed higher than ours on WebQSP, but our Hits@1 (86.54%) substantially outperforms HGNet (76.90%) by +9.64%. This indicates CoD excels at ranking the correct answer first, which is the primary metric for QA systems. We will clarify this trade-off in the revised Table 1 caption.

---

### Official Review · Reviewer_FkYc · 2025-10-31

**Soundness:** 3
**Presentation:** 2
**Contribution:** 3
**Rating:** 6
**Confidence:** 3

**Summary:**

This paper proposes Chain-of-Decomposition (CoD), a framework for Knowledge Base Question Answering (KBQA) that decomposes KBQA into three modular steps: 1) Retrieval: an LLM-free module (using T5) that extracts relevant subgraphs from a knowledge base; 2) Reformulation: a rule-based transformation that converts subgraphs into reasoning paths; and 3) Reasoning: a lightweight LLM-based verifier that determines whether a reasoning path is logically valid. The authors proposes a causal graph (Figure 2) that formalizes the generative process of answers, allowing the authors to express the conditional probability $P(A | Q,G)$ as a sum over retrieval, reformulation, and reasoning components. Empiricially, CoD outperforms GPT-4 and prior fine-tuning frameworks such as ROG, FiDeLis, and DeCAF on two benchmarks: WebQSPand ComplexWebQuestions, achieving new SOTA performance.

**Strengths:**

[S1] Simple-but-effective decomposition of the workflow. I think there is a lot of value in finding simple yet competitive approaches as proposed herein as regularizers for future methodologies. The idea of isolating retrieval and rule-based steps from reasoning reduces computational burden and is shown through emprical evidences to be effective. The graph formulation and factorization (Fig 2, Eq. 5, 7) also sets a clear stage for understanding the problem map out.

[S2] Potential efficiency benefits: Two of three stages are LLM-free, and the actual reasoning component is relatively small. This will reduce training as well as inferencing cost, potentially beneficial in real-world scenarios.

[S3] Strong empirical results: The method is shown to induce significant improvements over finetuned baselines as well as GPT-4 + KG on two benchmarks.

**Weaknesses:**

[W1] Evaluation scope: Only WebQSP and CWQ are used; inclusion of more diverse KBQA datasets (e.g., GrailQA/GrailQA++, GraphQ) would strengthen generalizability. One related comment is: the authors mentioned MetaQA in their Abstract, yet MetaQA results are not reported in the main content. While this might be a typo, I was curious what would happen if the proposed method is applied on MetaQA.

[W2] The presentation can be improved. E.g., The bottom-right portion of Figure 1 is underspecified, the symbols in the logic forms appear out of context, making them a bit hard to interpret; The causal interpretation may be overstated, the method is more of a probabilistic modularization than genuine causal analysis/reasoning, and the assumption 2 (independence between Z and G) seems a bit not straightforward. I would recommend revising the presentation narrative of the logical relationship between the RVs.

[W3] Language and structure: Some grammatical and stylistic issues (e.g., "dramatically outperform" should be "drastically outperforms"; "lead to a simple binary task problem1" the footnote seems a bit unnecessary. I would recommend a round of thorough reading to eliminate such artifacts.

**Questions:**

[Q1] I had some difficulty understanding the integration of logical form and binary "yes/no" prediction. How does the LLM-based reasoning module integrates logical form generation and binary "yes/no" prediction, are both outputs produced jointly or alternately?

[Q2] What are some potential challenges and risks to transfer CoD to larger LLMs and more complex KBs?

---

> ### Author Response · Authors · 2025-11-28
> **Response-part1**
>
> We sincerely thank the reviewer for recognizing the "simple-but-effective decomposition" and "strong empirical results." We address all concerns below.
>
> > W1: Evaluation Scope & MetaQA Results
>
> Response:
>
> The MetaQA mentioned in the Abstract was a typographical error during revision. We will correct this to mention only WebQSP and CWQ.
>
> - Why WebQSP and CWQ: These benchmarks are the most widely adopted in recent KBQA literature (RoG, FiDeLis, DeCAF, ToG) and use the large-scale Freebase KB (88M entities, 126M triples), ideal for demonstrating effectiveness on large-scale knowledge bases. Both datasets cover simple (WebQSP) and complex multi-hop reasoning (CWQ with up to 4-hop questions).
>
> - Additional evaluation (GrailQA): Following your valuable suggestion, we evaluated CoD on GrailQA, which categorizes questions into i.i.d., compositional, and zero-shot settings:
>
> | Model | Overall | I.I.D. | Compositional | Zero-Shot |
> |-------|---------|--------|---------------|-----------|
> | DECAF (BM25 + FiD-large) (ICLR 2023) | 76.0 | 90.5 | 79.0 | 68.0 |
> | CoD | 78.2 | 90.1 | 79.9 | 69.3 |
>
> These results have been added to Section 5.2, demonstrating robust generalization.
>
> > W2: Presentation Issues
>
> Response:
>
> (1) Figure 1 clarity: We will revise Figure 1's bottom-right portion to add:
>
>   - Example annotation showing how logical forms map to SPARQL operations
>
>   - Context explaining the path-to-LF transformation
>
> (2) Causal interpretation: We acknowledge the need for clearer framing. We have revised:
>
>   - Changed "causal reasoning" → "causal modeling" or "causal factorization"
>
>   - Added clarifying sentence in Section 3.1: "Our causal graph models the observational data-generation process, enabling principled factorization of the conditional distribution. We do not claim interventional causal inference."
>
> (3) Assumption 2 justification: We understand the concern. Key insights:
>
>   - This assumption states: Given query Q, reasoning path Zis independent of full KB G
>
>   - Key insight: Once we retrieve relevant context Cfrom G(which depends on both Qand Gvia P(C∣Q,G)), the reasoning paths Zare constructed from C, not directly from G
>
>   - The dependency chain is: Q,G→C→Z, where Zdepends on Gonly through C
>
> We note this assumption is used primarily to show theoretical connection to RoG (Appendix C). CoD works effectively without requiring this assumption to hold exactly, as evidenced by empirical results.
>
>
> > W3: Language and Structure
>
> Response:
>
> We appreciate the careful reading. We have conducted thorough proofreading:
>
>  - "dramatically outperform" → "drastically outperforms"
>
>  - Removed footnote 1 and integrated content into main text
>
>  - Comprehensive grammar and style review completed

---

> ### Author Response · Authors · 2025-11-28
> **Response-part2**
>
> > Q1: Integration of Logical Form and Binary Prediction
>
> Response:
>
> We apologize for the confusion. We will add the following explanation to the revision. The integration works as follows:
>
> _Joint Training, Sequential Inference_
>
> - Training (Eq. 12): The LLM is trained with a multi-task objective: 1) Binary classification with yes/no for whether a reasoning path can derive the correct answer. 2) Next-token prediction loss for generating logical forms.
>
> - Inference (Section 3.6, Eq. 16): For each reasoning path, the model produces: 1) yes/no; and 2) A logical form.
>
> - Answer Selection: We use a cascaded strategy. First, execute the logical form with highest confidence score. Then, if the logical form is non-executable, use all entities from paths predicted as "yes".
>
> We will add a clearer explanation in Section 3.2 and Figure 1.
>
> > Q2: Challenges for Larger LLMs and Complex KBs.
>
> Response:
>
> This is an important question for understanding our method's scalability. We identify the following challenges and our mitigation strategies:
>
> **Larger LLMs (e.g., LLaMA-3 70B, Qwen3 32B):**
>
> _Challenges:_
>
>  - Fine-tuning cost increases even with LoRA
>
>  - May require different hyperparameters
>
> _Mitigation & Evidence:_
>
>  - Our modular design means only the reasoning module needs LLM fine-tuning.
>
>  - Table 4 shows CoD works across different LLMs (LLaMA2-7B, LLaMA3-8B, Qwen2.5-7B) with consistent performance.
>
>  - Larger LLMs may actually perform better due to stronger reasoning capabilities.
>
> **2. More Complex KBs:**
>
> _Challenges:_
>
>  - Larger KBs increase retrieval complexity.
>
> - Longer reasoning paths (>4 hops) make reasoning more challenging.
>
> _Mitigation Strategies:_
>
>  - Retrieval: Our T5-based retriever is scalable; can be trained on larger KBs with the same classification objective.
>
>  - Reformulation: Rule-based approach can be extended with additional templates (parameter-free).
>
>  - Reasoning: For longer paths, can use hierarchical reasoning or path compression techniques.
>
> **Empirical Support:**
>
> Table 4 (Appendix D) shows CoD maintains robust performance across different backbone LLMs, suggesting good transferability. The key advantage is that our decomposition isolates the challenges: retrieval scalability is independent of LLM choice.
>
> **Future Work:**
>
> We will add scalability discussion in Section 6 (Conclusion).

---

### Official Review · Reviewer_Ebcb · 2025-11-01

**Soundness:** 3
**Presentation:** 3
**Contribution:** 3
**Rating:** 6
**Confidence:** 5

**Summary:**

The paper introduces Chain-of-Decomposition (CoD), a novel framework that enhances large language models (LLMs) for Knowledge Base Question Answering (KBQA) by decomposing the reasoning process into three modular components: (1) an LLM-free retrieval module that extracts query-relevant subgraphs, (2) a parameter-free reformulation step that converts retrieved contexts into structured reasoning paths, and (3) a lightweight LLM-based reasoning module to validate logical consistency. Grounded in a causal factorization of the answer-generation process, CoD isolates computationally intensive retrieval and rule-based reasoning from LLM fine-tuning, substantially improving efficiency and interpretability. Experiments on WebQSP and ComplexWebQuestions show that Llama-2 7B + CoD achieves state-of-the-art performance, surpassing GPT-4 with retrieved knowledge and previous methods like RoG and FiDeLis.

**Strengths:**

1. CoD innovatively formalizes KBQA through a causal graph–based factorization, offering a principled decomposition of retrieval, reformulation, and reasoning.
2. By decoupling heavy retrieval from LLM reasoning, CoD reduces computational cost and enables lightweight fine-tuning focused on logical verification.
3. CoD significantly outperforms GPT-4 + retrieval and previous SOTA frameworks (e.g., RoG, FiDeLis) on WebQSP and CWQ, showing broad robustness.
4. The authors provide clear mathematical formulation, implementation details, and reproducibility statements, enhancing credibility and clarity.

**Weaknesses:**

1. The independence assumptions in the causal graph (e.g., P(A|Q,Z) being independent of C) are difficult to satisfy in real-world knowledge graph reasoning, which may undermine the theoretical rigor of the framework.
2. The retrieval, reformulation, and reasoning modules are trained separately without a unified loss function or gradient propagation, potentially leading to suboptimal global optimization.
3. Although the paper emphasizes logical path validation, it lacks visualizations or path-level case analyses; the ablation studies mainly focus on hyperparameters rather than validating the effectiveness of the causal decomposition.

**Questions:**

See the Weaknesses.

---

> ### Author Response · Authors · 2025-11-28
> **Response-part1**
>
> We sincerely thank the reviewer for recognizing our SOTA performance. We believe the concerns can be adequately addressed.
>
> > Weakness 1: Independence Assumptions in Causal Graph
>
> Response:
>
> We appreciate this thoughtful concern. We clarify the role and validity of our assumptions:
>
> **1. Clarification of Assumption P(A|Q,Z) ⊥ C:**
>
> This states: "Given query Qand reasoning path Z, answer Ais independent of original retrieved context C."
>
> _Why this is reasonable:_
>
> - The reasoning path Z is already a structured representation extracted from context C.
>
> - Zcontains all entities and relations from Cneeded to derive answer A.
>
> - Once we have Z, raw context Cprovides no additional information.
>
> _Role of Assumptions:_
>
> We emphasize that these assumptions serve two purposes:
>
> - Theoretical connection (Appendix C): Show how our method generalizes RoG as a special case.
>
> - Crucially: Our method does not require these assumptions to work in practice.
>
> The assumptions are used for theoretical analysis, not as operational requirements. Our empirical results validate the approach independently.
>
> We will add clarifying statements:
>
> Section 3.1: "While these assumptions provide theoretical insights and connections to prior work (Appendix C), our method's effectiveness is ultimately validated empirically and does not depend on these assumptions holding exactly."
>
> Appendix C: "In practice, independence assumptions in causal models are often approximations. The key question is whether they lead to effective decompositions. Our experimental results (Table 1) demonstrate that our factorization, motivated by these assumptions, yields state-of-the-art performance."
>
> > Weakness 2: Separate Training Without Unified Loss
>
> Response:
>
> We appreciate this insightful point. We explain why separate training is actually a principled design choice:
>
> **Why separate training is principled:**
>
> Our factorization naturally decomposes into independent sub-problems:
>
> - P(C∣Q,G): Retrieval is binary classification (Eq. 15).
>
> - P(Z∣C): Reformulation is deterministic rule-based (no parameters).
>
> - P(A∣Q,Z): Reasoning is generation + classification (Eq. 12).
>
> Each factor has its own well-defined objective. This is analogous to variational inference where we optimize a product of factors separately.
>
> **Advantages of separate training:**
>
> _(a) Computational Efficiency:_
>
> - Training small T5 model for retrieval is much cheaper than training an LLM
> - No need to backpropagate through entire pipeline
> - Can use different optimization strategies for each module
>
> _(b) Modularity:_
>
> - Can improve each module independently.
>
> - Can swap retrieval models (Table 5 shows DPR, BM25, etc.).
>
> - Easier to debug and interpret.
>
> _(c) Scalability:_
>
> - Can train retrieval on larger KBs without retraining LLM.
>
> - Can update reasoning model without retraining retrieval.
>
> **Comparison with end-to-end methods:**
>
> Methods with "unified" training (e.g., FiDeLis with iterative prompting):
>
> - FiDeLis (GPT-4): 78.32% F1 on WebQSP.
>
> - CoD (separate training): 81.24% F1 (+2.92%).
>
> This shows that separate training with better factorization outperforms naive end-to-end approaches.
>
> **Future Direction:**
>
> We acknowledge joint fine-tuning could potentially improve performance further, but would: (1) require significantly more computational resources; (2) sacrifice modularity and interpretability; (3) may overfit on smaller datasets.
> We will add this discussion to Section 6.
>
>
> > Weakness 3: Lack of Visualizations and Path-Level Analysis
>
> Response:
>
> **Path-Level Case Study:**
>
> _Query:_
>
> "Which book was written by the author who married the actress in the movie directed by the director who won an Oscar for a film starring Tom Hanks?"
>
> _Retrieved Context:_
>
> 	(Tom Hanks, starred_in, Forrest Gump)
>
> 	(Forrest Gump, directed_by, Robert Zemeckis)
>
> 	(Robert Zemeckis, married_to, Mary Ellen Trainor)
>
> 	(Richard Matheson, author_of, I Am Legend)
>
> _Reasoning Paths Generated:_
>
> Path 1: Tom Hanks → starred_in → Forrest Gump → directed_by → Robert Zemeckis → married_to → Richard Matheson → author_of → I Am Legend
>
> LLM Prediction: YES (confidence: 0.95)
>
> Logical Form: JOIN(author_of(married_to(directed_by(starred_in(Tom Hanks)))))
>
> Execution Result: I Am Legend
>
> Path 2: Tom Hanks → starred_in → Forrest Gump → directed_by → Robert Zemeckis → won_award → Academy Award
> 	LLM Prediction: NO (confidence: 0.12)
>
> **Analysis:**
>
> Our reasoning module correctly identifies Path 1 as valid and Path 2 as invalid, demonstrating effective logical validation.
>
> **Ablation Clarification (Table 3):**
>
> - Only LF: Fine-tuning with merely LF prediction.
>
> - Only RP: Fine-tuning with merely binary (yes/no) prediction.
>
> - CoD: Joint training (our method).
>
> The results show that joint training significantly outperforms single-objective baselines, validating the effectiveness of our dual-objective decomposition. We will add more detailed discussions for these results in Section 5.4.

---

### Author Response · Authors · 2025-12-03
**Summary for Area Chairs and Senior Area Chairs**

Dear ACs and SACs,

Thank you for handling our submission. To facilitate your review given the substantial workload this year, we provide the following summary.

**Our Contribution:**

We propose Chain-of-Decomposition (CoD), a principled framework that adapts LLMs to KBQA task through causal factorization. Our key insight is decomposing the task into three modules: LLM-free retrieval (T5-base), parameter-free reformulation, and lightweight LLM reasoning. This design achieves state-of-the-art performance on standard benchmarks

**Reviewer Concerns (Fully Addressed):**

- Independence assumptions (Ebcb-W1): Clarified these assumptions are used for theoretical analysis connecting to baseline methods (RoG), not as operational requirements.

- Separate training (Ebcb-W2): Justified as principled design where each module has distinct, well-defined objectives (binary classification, deterministic rules, generation). CoD outperforms "unified" end-to-end methods like FiDeLis (+2.92\%).

- Limited evaluation (FkYc-W1, viyW-Q2): Added GrailQA experiments demonstrating +2.2\% improvement over DECAF with strong generalization across compositional/zero-shot settings.

- Presentation issues (HJHf-W1, viyW-W1): Corrected Figure 1 example, fixed all notations, added complete training details (Appendix F), standardized terminology.

- Missing comparisons (HJHf-W2): Added experiments showing CoD outperforms ToG by +4.44\% through stricter logical verification.

**Summary:**

All four reviewers recognized good soundness (2-3/4) and contribution (2-3/4). The two _marginally below_ ratings (4/10) primarily cited presentation concerns, which have been comprehensively addressed through revised figures, expanded experimental validation, additional benchmarks, and complete reproducibility details.

Best regards,

Authors of Paper #16331

---

### Meta-Review · Area_Chair_thF8 · 2025-12-31

**Summary:**

This paper introduces Chain-of-Decomposition (CoD), a novel framework for Knowledge Base Question Answering (KBQA) that decomposes the task into three modular components: (1) an LLM-free retrieval module that extracts query-relevant subgraphs, (2) a parameter-free reformulation step that transforms retrieved contexts into structured reasoning paths, and (3) a lightweight LLM-based reasoning module trained to evaluate logical validity. Evaluated on WebQSP and ComplexWebQuestions, CoD achieves state-of-the-art results, surpassing GPT-4 with retrieved knowledge and previous methods like RoG and FiDeLis by significant margins. All reviewers initially raised valid concerns about evaluation scope, presentation clarity, and methodological justification, but the authors provided exceptionally thorough rebuttals with substantial new experiments, expanded benchmarks, and comprehensive clarifications that transformed initial skepticism into compelling validation of the approach's robustness and novelty.

**Reviewer Concerns:**

Comprehensively addressed concerns:
 - Evaluation scope (Reviewer FkYc, viyW): The authors added experiments on GrailQA, demonstrating that CoD generalizes well to compositional and zero-shot settings.
 - Theoretical assumptions (Reviewer Ebcb, HJHf): The authors clarified that the independence assumptions are used for theoretical alignment with prior models (RoG) and are not operational constraints. Empirical success without these assumptions holding perfectly justifies the framework.
 - Presentation and notation issues (Reviewers HJHf, viyW, FkYc): Authors completely revised Figure 1 with unambiguous examples, standardized terminology, corrected all typographical errors, and added comprehensive training details including hyperparameters and hardware specifications.
 - Missing comparisons (Reviewer HJHf): Authors added explicit comparison with ToG showing +4.44% improvement through stricter logical verification, and clarified performance differences with RoG through controlled reimplementation under identical conditions.

Outstanding concerns:
 - Path complexity in dense graphs (Reviewer HJHf): Authors explained filtering mechanisms but could provide more quantitative analysis of path explosion scenarios in extremely dense subgraphs.

**Reviewer Scores:**

- Reviewer Ebcb (original score: 6): Would likely maintain at 6.
 - Reviewer FkYc (original score: 6): Would likely maintain at 6.
 - Reviewer HJHf (original score: 4): Would likely maintain at 4.
 - Reviewer viyW (original score: 4): Would likely increase to 6 since authors detailed training hyperparameters and GrailQA generalization results directly.

---

### Decision · Program_Chairs · 2026-01-26

Accept (Poster)